# SimpleTrack: Rethinking and Improving the JDE Approach for Multi-Object Tracking

**DOI:** 10.3390/s22155863

**Published:** 2022-08-05

**Authors:** Jiaxin Li, Yan Ding, Hua-Liang Wei, Yutong Zhang, Wenxiang Lin

**Affiliations:** 1Key Laboratory of Dynamics and Control of Flight Vehicle, Ministry of Education, School of Aerospace Engineering, Beijing Institute of Technology, Beijing 100081, China; 2Department of Automatic Control and Systems Engineering, University of Sheffield, Sheffield S1 3JD, UK

**Keywords:** multiple object tracking, association matrix, joint detection and embedding, decoupling representation

## Abstract

Joint detection and embedding (JDE) methods usually fuse the target motion information and appearance information as the data association matrix, which could fail when the target is briefly lost or blocked in multi-object tracking (MOT). In this paper, we aim to solve this problem by proposing a novel association matrix, the Embedding and GioU (EG) matrix, which combines the embedding cosine distance and GioU distance of objects. To improve the performance of data association, we develop a simple, effective, bottom-up fusion tracker for re-identity features, named SimpleTrack, and propose a new tracking strategy which can mitigate the loss of detection targets. To show the effectiveness of the proposed method, experiments are carried out using five different state-of-the-art JDE-based methods. The results show that by simply replacing the original association matrix with our EG matrix, we can achieve significant improvements in IDF1, HOTA and IDsw metrics, and increase the tracking speed of these methods by around 20%. In addition, our SimpleTrack has the best data association capability among the JDE-based methods, e.g., 61.6 HOTA and 76.3 IDF1, on the test set of MOT17 with 23 FPS running speed on a single GTX2080Ti GPU.

## 1. Introduction

Multi-object tracking (MOT), aiming to estimate the locations and identity of multiple targets in a video sequence, is a fundamentally challenging task in computer vision [1]. Recently, the Intersection over Union (IoU) and Hungarian method have been commonly used in the tracking phase, among many tracking-by-detection paradigms [2,3,4,5,6,7,8,9,10]. However, when the target is occluded or lost for a period of time, it is difficult to retrieve the correct identity only using the IoU distance. As a result, the identity switching of targets occurs from time to time. To alleviate this problem, many methods have started to introduce the re-identity feature of targets. Among them, the JDE-based methods [11,12,13,14,15,16,17] have become popular due to their simplicity and efficiency.

In part of the data association, the accuracy of similarity measurement determines the tracking performance. Most detection-based methods use the IoU distance as the similarity matrix in the cascade matching strategy, while JDE-based methods fuse the motion information and appearance information as the similarity matrix for the linear assignment in the first matching and use the IoU distance in the next matching. However, none of these existing methods provides the best expression of the similarity matrix according to our experiments.

When objects are occluded due to interlacing, it will produce confusing sets, which are difficult to allocate correctly, e.g., the set {det4, det7, track4, track10} in Figure 1a,b, and the set {det4, det7, track4, track9} in Figure 1c. When assigning these confusing sets, the inaccurate similarity distance leads to tracking failure. Based on the Hungarian method, the IoU distance matrix tends to match det4 with track4 and det7 with track10, and the EM distance matrix tends to match det4 with track4 and det7 with track9. Both of them lead to target identity switching. The principal reason for these matching failures is the inaccurate prediction from the Kalman filter as the time of target loss becomes longer. Clearly, this results in an inaccurate IoU distance and motion information distance, which leads to the problem of linear allocation errors.

To solve this problem, we propose the EG matrix, which utilizes the embedding cosine distance for the long-range tracking of targets and the GioU distance for limiting the matching range of embedding. To illustrate the robustness of the EG matrix, we apply it to five different JDE-based methods. As can be seen in Section 4.3, our implementations obtain improvements in MOT metrics, including tracking speed, HOTA, IDF1 and IDsw metrics.

To further explore the good properties of the EG matrix, we propose a simple tracking framework named SimpleTrack. In this framework, we design a bottom-up branch to represent Re-ID features. Different from the fusion method of detection features, it pays more attention to the high-level semantic layers. For the tracking part of SimpleTrack, we propose a novel tracking retrieval mechanism and design a new tracking strategy based on our EG matrix. The experimental results show that our tracking strategy can surpass the JDE-based methods in most metrics, including tracking speed. Compared with the current SOTA method BYTE, our tracking strategy can also improve the performance in terms of HOTA, IDF1 and IDsw metrics.

Our main contributions are as follows:

1. We adopt different feature fusion structures for feature detection and feature re-identification, respectively, to decouple them.

2. We propose a novel association matrix named the embedding and GioU matrix, which can directly replace the original association matrix in JDE-based methods. It can not only reduce time costs, but also improves the tracking metrics of the model.

3. We design a new tracking strategy that can alleviate the problem of tracking target loss.

4. The code and model are available at https://github.com/1vpmaster/SimpleTrack (accessed on 3 July 2022).

The remainder of the paper is arranged as follows. Section 2 summarizes the related work, including JDE-based methods, similarity matrices and tracking strategies. Section 3 describes the method of SimpleTrack, including the decoupling module, embedding and Giou matrix and a novel tracking strategy. In Section 4, experimental results are provided to verify the performance of the proposed SimpleTrack. Section 5 discusses the performance and speed of the EG matrix and association methods. Section 6 briefly summarizes the work and considers the future work.

## 2. Related Work

### 2.1. Joint Detection and Embedding

JDE-based methods typically employ a single network to directly predict detection and appearance features [11,12,13,14,15,16,17,18,19]. In general, these methods employ a single backbone to predict both object bounding boxes and appearance features. For example, FasterVideo [18] and Online Tracker [19] adopt Faster R-CNN [20] and Yolov5 for feature detection and feature re-identification, respectively. Although their pipelines are relatively simple, the competitive relationship between detection and identification harms the optimization procedure in the multi-task learning of object detection and appearance feature extraction.

Recently, to tackle this problem, CSTrack [13] was proposed, which first uses a decoupling module to enhance the learned representation for both object detection and appearance identification. RelationTrack [21] uses a channel attention mechanism to decouple detection and re-identity. However, the two methods do not take into account the essential differences between detection features and re-identity features. Different from CSTrack and RelationTrack, the decoupling strategy adopted in our SimpleTrack focuses on the essence of the appearance feature. We start decoupling from the feature layer fusion of the network. In contrast to the detection feature fusion, we adopt a bottom-up fusion method.

### 2.2. Similarity Matrices

Location, motion and appearance are the most common cues in multi-object tracking. They are also combined together for the linear assignment. Detection-based methods [10] utilize the IoU distance as the similarity matrix and the tracking accuracy mainly depends on the detector. SORT [2] fuses position and motion cues as the similarity matrix, which can achieve good results in short-range matching. DeepSORT [7] improves the long-range tracking ability of trackers by merging appearance and motion cues, which is usually used in JDE-based methods [11,12,13,14,15,16,17].

All these methods use location cues or fuse appearance and motion information as the similarity matrix, as shown in Figure 2. However, the motion information estimated by linear motion models is not accurate in some scenes containing complex motion behaviors. In addition, it is time-consuming to integrate motion information and appearance information according to Section 5.2.2. Different from all the aforementioned methods, we design the similarity matrix combined with appearance and location information and use the GioU distance matrix as the location cue instead of the common IoU matrix.

### 2.3. Tracking Strategy

The assignment problem of target tracking and detection can be solved by the Hungarian algorithm [22] based on different similarity matrices. SORT associates the detection objects with the tracking objects by one-time matching. DeepSORT adopts a cascade matching method that reduces unmatched tracking targets. MOTDT [23] first uses the appearance similarity matrix and the IoU distance matrix as the similarity matrix for cascade matching, respectively. All of these methods assume that the detection targets are equally important and match them uniformly with the similarity matrix.

Recently, BYTETrack [10] proposed to use low-confidence detection results for secondary matching, which reduces the problem of target detection failure due to occlusion. Thereby, the occurrence of long-range tracking could be reduced, making the linear assignment based on the IoU distance matrix more effective. MAA [24] adopts different strategies for the blurred detection of targets and tracking targets in the similarity matrix. The method can alleviate the inaccuracy of the similarity distance caused by the ambiguous targets. Both of the two methods aim to make up for the shortcomings of the similarity matrix and do not pay attention to how to retrieve the lost detection targets. Based on the idea of BYTE [10], we redesign the similarity matrix for the JDE-based method and construct a new matching strategy.

## 3. SimpleTrack

In this section, we present the technical details of SimpleTrack, as illustrated in Figure 3. It is composed of feature decoupling, a similarity matrix as well as a tracking strategy.

### 3.1. Feature Decoupling

We adopt DLA-34 as a backbone in order to strike a good balance between accuracy and speed. For feature decoupling, we employ different feature fusion methods for detection and Re-ID representation. As illustrated in Figure 3, for the detection branch, the feature fusion method still adopts the structure of IDA-up in FairMOT [12]. We call it the up-to-bottom fusion method, based on low-level feature maps and continuously fusing higher-level feature maps.

However, Re-ID features tend to learn higher-level semantic features to distinguish different features among homogeneous objects. Therefore, we take a simple bottom-up approach to fusing feature maps. Denote the input feature maps by $F={\left\{F_{i}\right\}}_{i=1}^{N}$, where *N* is the number of feature layers of different resolutions extracted by the backbone network. Then, the process of the bottom-up fusion method can be expressed as
(1)$$\begin{array}{c} {\left\{{\hat{F}}_{i}\right\}}_{i=N}^{1}=\left\{\begin{array}{cc} F_{i}, & \mathrm{if}\;i=N\\ F_{i}\cdot \sigma \left(Conv_{1\times 1}\left(UpSample\left({\hat{F}}_{i+1}\right)\right)\right), & \mathrm{otherwise} \end{array}\right. \end{array}$$
where UpSample(·) represents an upsampling operation composed of the deformable convolution and the deconvolution, $Conv_{1\times 1}$ denotes a 1×1 convolution layer for changing channels of features, σ(·) represents the Sigmoid activation layer.

It could be observed from Equation (1) that the fusion process is from bottom to top, and the previously fused feature map guides the lower-level feature map until the final fusion result is obtained. As will be shown by the experimental results in Section 4, the computational cost required by this fusion method is minimal.

### 3.2. Embedding and GioU Matrix

The similarity matrix is usually constructed from location, motion and appearance information. Let L, M, E denote the location distance matrix, the motion distance matrix and the appearance distance matrix, respectively. We fuse L and E as the similarity matrix, called the EG matrix. Moreover, L can be represented as
(2)$$\begin{array}{c} L=1-(\frac{|A\cap B|}{|A\cup B|}-\frac{|C\setminus (A\cup (B))|}{|C|}) \end{array}$$
where A and B represent the bounding boxes of the tracking objects and the bounding boxes of the detection objects, respectively, and C is the minimum enclosing rectangle sets of the above bounding boxes.

E can be represented as
(3)$$\begin{array}{c} E=\frac{O_{e}^{1}\cdot O_{e}^{2}}{\|O_{e}^{1}\left\|\right\|O_{e}^{2}\|} \end{array}$$
where $O_{e}^{1}$ and $O_{e}^{2}$ represent different appearance embedding vectors.

Note that the matrix L in Equation (2) is actually the GioU distance matrix and that the matrix E in Equation (3) defines the cosine distance matrix. Then, the embedding and GioU matrix, which is also denoted as EG, can be represented as
(4)$$\begin{array}{c} \mathrm{EG}=\lambda_{1}E+\lambda_{2}G \end{array}$$
where $\lambda_{1}=1.0$ and $\lambda_{2}=0.5$ represent two hyperparameters, G denotes the GioU distance matrix and G=L.

### 3.3. Tracking Strategy in SimpleTrack

Inspired by BYTE [10], we develop a tracking strategy based on our EG matrix. As shown in Algorithm 1, we follow the idea of secondary matching with low-confidence detection adopted in BYTE, and use the EG matrix to replace the similarity matrix in the cascade matching. In addition, after the secondary matching, we utilize the cosine distance to retrieve the unmatched tracklets.

As shown in Figure 4, when the target is blocked and the detector fails, we use a Kalman filter to predict the center point position of the unmatched tracking targets. In order to compensate for the drift of the Kalman filter, we use appearance information to modify the prediction results of the Kalman filter. We select the appearance embedding vectors in the 3 × 3 range around the prediction center point $\left(C_{i},C_{j}\right)$ by the Kalman filter. Denoting the embedding vector of the unmatched tracking target by $E_{u}$, we follow Equation (5) to determine whether the unmatched tracking target can be retrieved.
(5)$$\begin{array}{c} S\left(t_{u}\right)=\left\{\begin{array}{cc} \mathrm{retrieved}, & \mathrm{if}\;Dis_{min}<\epsilon_{r}\\ \mathrm{unretrieved}, & \mathrm{otherwise} \end{array}\right. \end{array}$$
where $Dis_{min}$ represents the minimum cosine distance among the 3 × 3 range around $\left(C_{i},C_{j}\right)$, $\epsilon_{r}$ denotes the tracking retrieval threshold. By denoting the appearance embedding vector on pixel (i,j) by $E_{d}^{\left(i,j\right)}$, $Dis_{min}$ can then be represented by Equation (6).
(6)$$\begin{array}{c} Dis_{min}=Min_{i,j}\left({\left\{F_{cd}\left(E_{d}^{i,j},E_{u}\right)\right\}}_{i\in \left[C_{i}-1,C_{i}+1\right],j\in \left[C_{j}-1,C_{j}+\right]}\right) \end{array}$$
where $F_{cd}\left(\cdot \right)$ indicates the results of the cosine distance between two vectors. Afterward, if the state of the unmatched tracking target is judged to be retrieved, we obtain $\left(i_{min},j_{min}\right)$ according to $Dis_{min}$. Finally, set $\left(i_{min},j_{min}\right)$ as the center point of the retrieval box, and make the width and height of the retrieval box consistent with the tracked target in the previous frame.

With the tracking retrieval mechanism, we can recover the occluded (failed) detection boxes by using the predictions of the Kalman filter. At the same time, the embedding information can be used to correct the predicted position of the Kalman filter, so as to update the parameters of the Kalman filter and reduce the accumulated error of the Kalman filter.
**Algorithm 1: **Pseudo-code of SimpleTrack
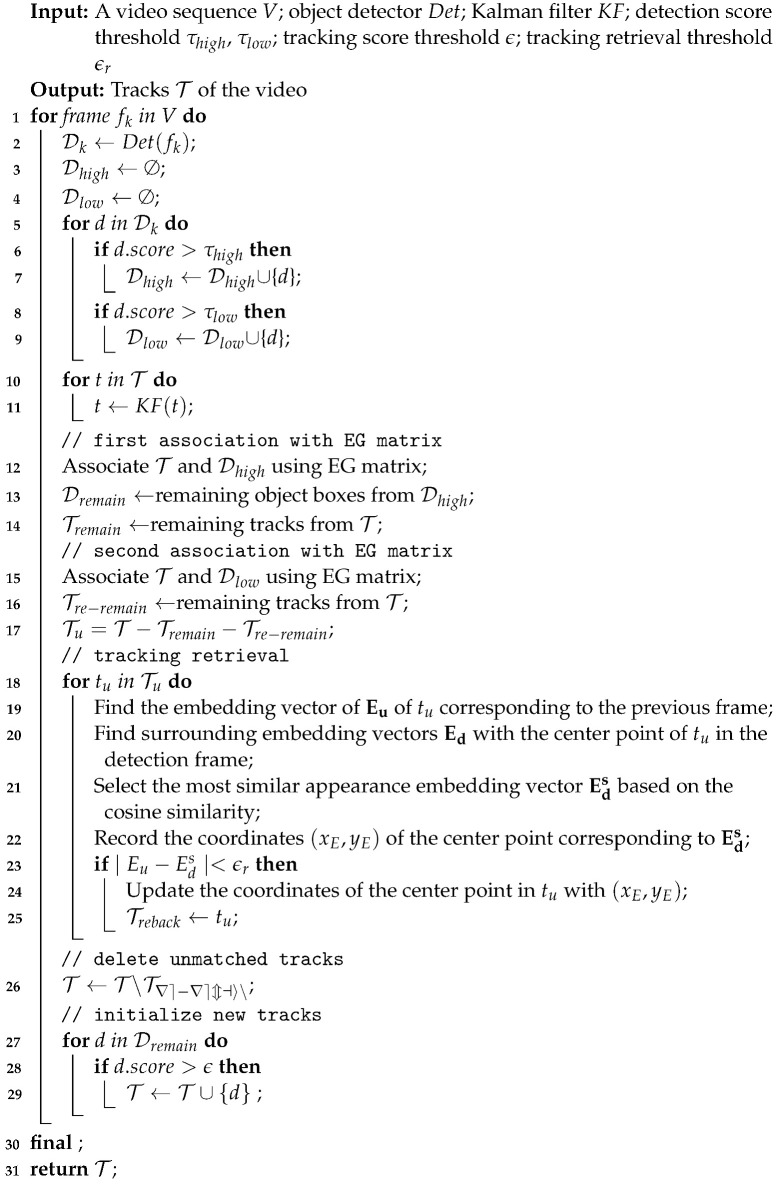


## 4. Experiments

### 4.1. Datasets and Metrics

#### 4.1.1. Datasets

We evaluate SimpleTrack on private detection tracks of the MOT17 [25] and MOT20 [26] datasets. The former contains 14 different video sequences for multi-target tracking, recorded by fixed or moving cameras. The latter consists of 8 video sequences with a fixed camera focusing on tracking in very crowded scenes, 4 for training and testing each. For ablation studies, we follow [27,28,29,30,31] and split the train set into two parts for ablative experiments as the annotations of the test split are not publicly available. We fuse the CrowdHuman [32] and MOT17, with half as the training dataset for ablation experiments following [10,30,31,33,34]. We add the ETH [35], CityPerson [36], CalTech [37], CUHK-SYSU [38] and PRW [39] datasets for training following [11,12,13] when testing on the test set of MOT17.

#### 4.1.2. Evaluation Metrics

To evaluate the tracking performance, we use TrackEval to evaluate all metrics, including MOTA [40], IDF1 [41], false positives (FP), false negatives (FN), identity switches (IDSW) and the recently proposed HOTA [42]. HOTA can comprehensively evaluate the performance of detection and data association. IDF1 focuses more on the association performance and MOTA evaluates the detector ability and focuses more on detection performance.

### 4.2. Implementation Details

#### 4.2.1. Tracker

In the tracking phase, the default high detection score threshold $\tau_{high}$ is 0.3, the low threshold $\tau_{low}$ is 0.2, the trajectory initialization score ϵ is 0.6, and the trajectory retrieval score $\epsilon_{r}$ is 0.1, unless otherwise specified. In the linear assignment step, for the high-confidence detection, the assignment threshold is 0.8, and for the low-confidence detection, the assignment threshold is 0.4.

#### 4.2.2. Detector and Embedding

We use SimpleTrack to extract the location features and appearance features of objects. For SimpleTrack, the backbone is DLA-34, which initializes weights with a COCO-pretrained model. The training schedule is 30 epochs on the combination of MOT17, CrowdHuman and other datasets mentioned above. The input image size is 1088 × 608. Rotation, scaling and color jittering are adopted as data augmentation techniques during our training phase. The model is trained on 4 NVIDIA TITAN RTX with a batch size of 32. The optimizer is Adam and the initial learning rate is set to $2\times 10^{-4}$, which decays to $2\times 10^{-5}$ in the 20th epoch. The total training time is around 25 h. FPS is measured with a single NVIDIA RTX2080Ti and the batch size is set to 1.

### 4.3. Ablation Studies

#### 4.3.1. Ablation on SimpleTrack

The innovation of SimpleTrack is mainly composed of bottom-up decoupling, the EG similarity matrix and tracking retrieval. We conduct ablation experiments on the MOT17 validation set for these three modules. The results are shown in Table 1. It can be observed that adding bottom-up decoupling to FairMOT increases IDF1 and MOTA. In addition, after replacing the similarity matrix of JDE-based methods with the EG matrix, the strategy improves IDF1 from 76.1 to 78.1, MOTA from 71.4 to 72.5 and HOTA from 60.2 to 61.5 and decreases IDs from 451 to 186. After further adding the tracking retrieval mechanism, the IDF1 metric increases from 78.1 to 78.5 and HOTA from 61.5 to 61.7, and the IDs metric decreases from 186 to 182. These results prove that the modules proposed in SimpleTrack are necessary and effective.

#### 4.3.2. Analysis of the Hyperparameters of EG Matrix

We test different sets of hyperparameters of Equation (Equation 4) in Figure 5. Set the parameter $\lambda_{1}$ of embedding similarity to 1 and increase the parameter $\lambda_{2}$ of GioU similarity from 0.1 to 4. It can be observed that the tracking performance of the algorithm is better when $\lambda_{1}$ is set to 1.0 and $\lambda_{2}$ is set from 0.5 to 0.9, because the interval of embedding similarity is [0, 1], and the interval of GioU similarity is [0, 2]. Therefore, in order to balance the weights of embedding similarity and GioU similarity, EG matrix set $\lambda_{1}=1.0$ and $\lambda_{2}=0.5$.

#### 4.3.3. Comparison with Preceding SOTAs

In this part, we compare the performance of SimpleTrack with preceding SOTA methods on MOT17 and MOT20. The results are reported in Table 2 and Table 3, respectively. As shown in these two tables, SimpleTrack showed the best results in various metrics and surpassed the contrasted counterparts by large margins, especially on the HOTA, IDF1 and IDS metrics. Moreover, compared with other MOT tracking methods, SimpleTrack has an obvious speed advantage.

#### 4.3.4. Visualization Results

We show some scenarios that are prone to identity switching in Figure 6, which contains three sequences from the half validation set of MOT17. We use different tracking strategies to generate the visualization results. It can be observed that SimpleTrack can effectively deal with the identity switching problem caused by the occlusion of the tracking targets. In addition, some tracking examples on the MOT17 test datasets are shown in Figure 7.

## 5. Discussion

This section mainly discusses the similarity matrix and the tracking association method. Section 5.1 analyzes and compares the performance of our proposed EG matrix with other existing similarity metrics, and applies the EG matrix to other JDE-based methods to analyze the universality of the EG matrix. Section 5.2 compares the speed and accuracy of our proposed tracking strategy with other existing tracking methods.

### 5.1. Analysis of the Similarity Matrix

#### 5.1.1. Performance Compared with Other Similarity Metrics

We employ different distance matrices as the similarity measure and evaluate their data association ability on the half validation set of MOT17. It can be obtained from Table 4 that only using the GioU or embedding matrix for data association does not result in good performance. Moreover, the table shows that the combination of the embedding matrix and IoU matrix can improve the association effect but reduces the result of MOTA. Compared with the IoU matrix used in detection-based methods, our EG matrix improves the IDF1 from 75.7 to 78.5 and HOTA from 60.4 to 61.7 and decreases IDs from 285 to 182. Compared with the embedding and motion matrix used in JDE-based methods, our EG matrix improves both the MOT metrics and tracking speed.

#### 5.1.2. Applications in Other JDE-Based Trackers

We apply our EG matrix to five different JDE-based trackers, including JDE [11], FairMOT [12], CSTrack [13], TraDes [30] and QuasiDense [16]. Among these trackers, JDE, FairMOT, CSTrack, TraDes merge the motion and Re-ID similarity and the first three methods follow the same fusion strategy. QuasiDense uses Re-ID similarity alone. It can be observed from Table 5 that using the EG matrix instead of the EM matrix can enhance the tracking performance and improve the tracking speed. Taking the JDE [11] method as an example, only using the EG matrix to replace the EM matrix can improve the HOTA from 50.1 to 50.9, IDF1 from 63 to 64.4, MOTA from 59.3 to 59.5 and FPS from 16.64 to 21.29 and decreases the IDs from 621 to 558. Combined with the BYTE strategy, our EG matrix still improves the HOTA from 50.4 to 50.9, IDF1 from 64.1 to 64.4 and FPS from 18.52 to 25.48 and decreases the IDs from 437 to 388.

### 5.2. Analysis of the Association Methods

#### 5.2.1. Accuracy Compared with Other Association Methods

We compare SimpleTrack with other association methods, including the recent SOTA algorithm BYTE and the tracking algorithm used in JDE-based methods [11,12,13,17]. As shown in Table 6, SimpleTrack improves the IDF1 metric of JDE from 76.1 to 78.5, MOTA from 71.4 to 72.5 and HOTA from 60.2 to 61.7 and decreases IDs from 451 to 182. Compared with BYTE, we can see that SimpleTrack improves the IDF1 from 75.7 to 78.5 and HOTA from 60.4 to 61.7, and decreases IDs from 285 to 182. These demonstrate that our tracking method is more effective than the JDE strategy, and it can improve the accuracy of data association compared to the BYTE strategy.

#### 5.2.2. Speed Compared with Other Association Methods

From Table 4 and Table 6, we can observe that our SimpleTrack algorithm utilizes the embedding information but is still nearly 20% faster than the JDE-based tracking strategy. A more detailed comparison of different video sequences can be observed in Figure 8a. It can be observed that our tracking algorithm is only slightly slower than BYTE, which does not utilize the embedding information. According to Figure 8b, we can see the time consumption of the main modules in the tracking phase. It shows that the JDE-based tracking strategy spends a lot of time in fusing the embedding and motion information, which is represented by the orange dotted square in Figure 8b. For the EG matrix, we only need to calculate the GioU distance and add it to the embedding distance. The time consumption is represented by the orange dotted star in Figure 8b.

## 6. Conclusions and Future Work

We propose a simple yet effective data association matrix, the EG matrix, for JDE-based multi-object tracking methods. The EG matrix can be easily applied to existing trackers and improves not only the tracking effect but also the speed of JDE-based tracking methods. In addition, we design a bottom-up feature fusion module for decoupling Re-ID and detection tasks, and present a novel tracklet retrieval strategy for mitigating the loss of detection targets. These innovations together form our SimpleTrack, which achieves 61.6 HOTA and 76.3 IDF1 on the test set of MOT17 with 23 FPS, ranking first among all the JDE-based methods.

SimpleTrack has a strong data association ability due to adopting the EG matrix and decoupling feature extraction module, which can be applied to multi-target tracking in some complex scenes. In the future work, we will consider the enhancement of target features in the time dimension and design an anti-occlusion feature extraction network based on our SimpleTrack framework. Moreover, we hope that the EG matrix can become the standard association matrix of JDE-based methods in multi-object tracking.

## Figures and Tables

**Figure 1 sensors-22-05863-f001:**
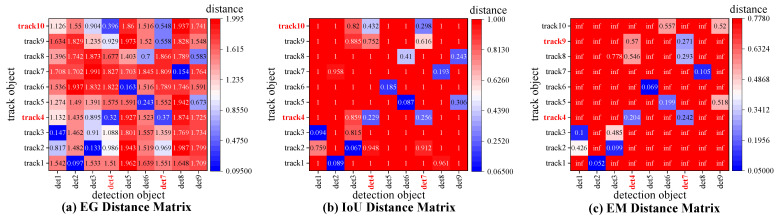
Example of heatmaps for different association matrices in frame 560 of MOT17 sequence 11. (**a**) shows our EG matrix, which combines the embedding cosine distance and the GioU distance. (**b**) shows the IoU distance matrix, i.e., the detection-based methods. (**c**) shows the EM matrix, which usually combines the motion distance and the embedding cosine distance, i.e., the JDE-based methods. In these heatmaps, the red cells indicates that the similarity distance between detection targets and tracking targets is farther, and the blue cells show that the similarity distance is closer.

**Figure 2 sensors-22-05863-f002:**
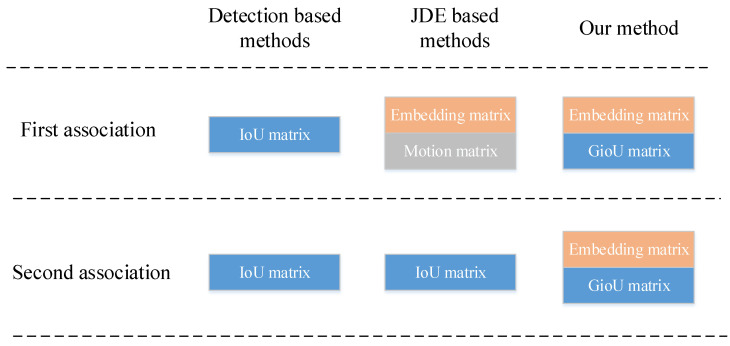
Association matrices used in cascade matching of different tracking methods.

**Figure 3 sensors-22-05863-f003:**
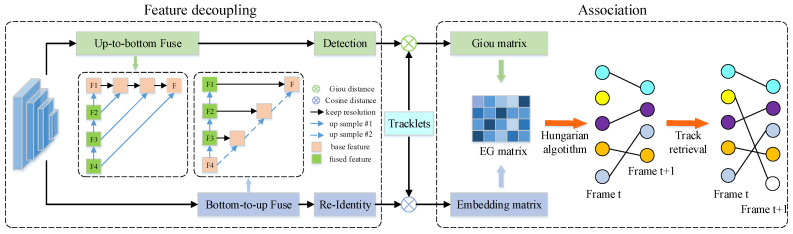
The overall pipeline of SimpleTrack. The input image is first fed to a backbone network to extract high-resolution feature maps. Then, we use different feature fusion methods for detection and re-identity separately, and combine the embedding and GioU distance matrix as the similarity matrix. At the end of the association phase, the tracking retrieval mechanism is used to recover the undetected targets.

**Figure 4 sensors-22-05863-f004:**
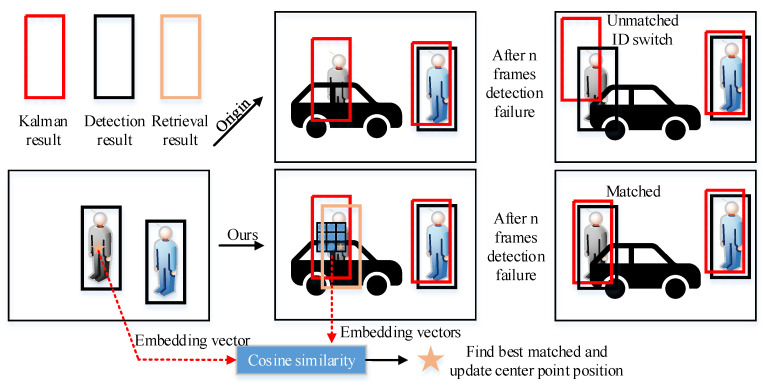
Tracking retrieval process. The five-pointed star indicates the position of the best matching embedding vector.

**Figure 5 sensors-22-05863-f005:**
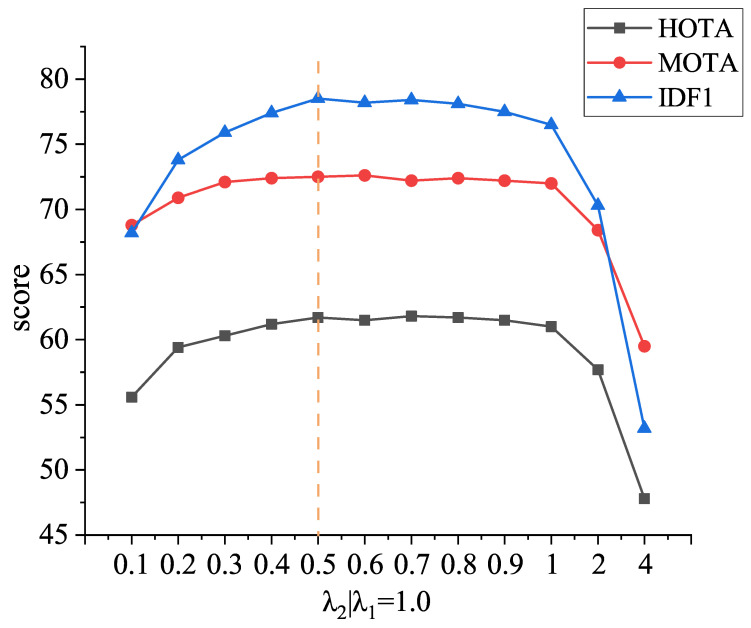
Experiments for hyperparameters in EG matrix in MOT17-half val. The blue, red and black lines represent IDF1, MOTA and HOTA indicators, respectively. The parameters selected in the paper are shown by the dotted line.

**Figure 6 sensors-22-05863-f006:**
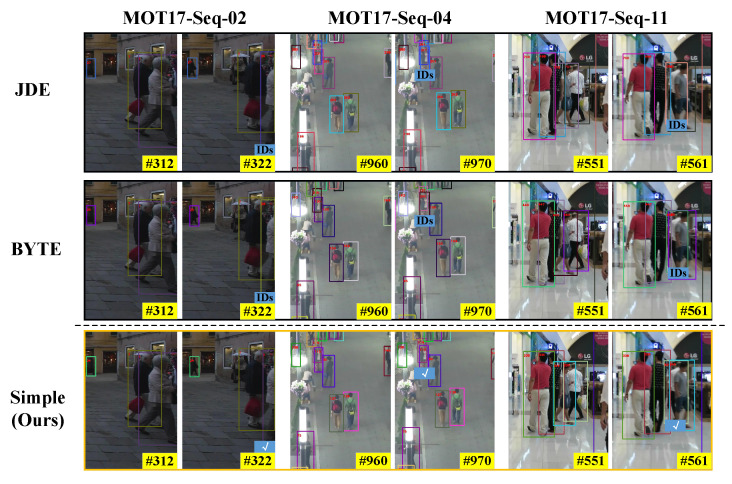
Robustness of our tracking strategy compared to BYTE and JDE-based methods. Boxes with the same color indicate that the tracking targets have the same identity; IDs indicates that the tracking targets have switched their identities. The check mark indicates that the identity of the target has not changed. The #number indicates that the frame number in mot17 video sequence.

**Figure 7 sensors-22-05863-f007:**
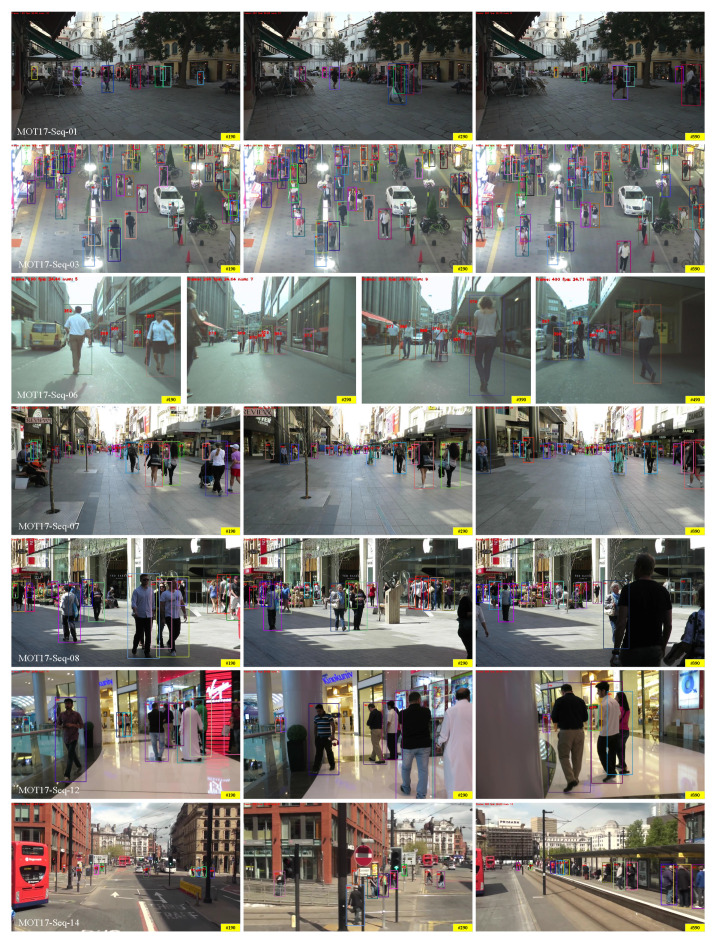
Tracking results of SimpleTrack on the MOT17 test dataset. The #number indicates that the frame number in mot17 video sequence.

**Figure 8 sensors-22-05863-f008:**
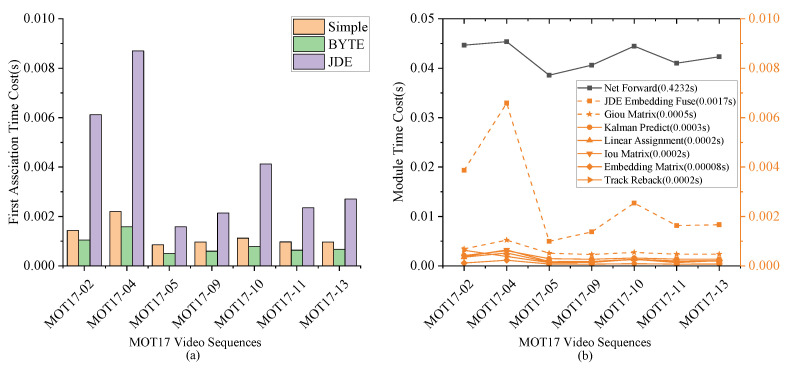
Comparison of different tracking algorithm speeds. (**a**) shows the tracking speed of different tracking algorithms. (**b**) shows the time consumption of several main modules in the tracking phase.

**Table 1 sensors-22-05863-t001:** Ablation experiment on SimpleTrack. ✔ denotes addition of this module to the baseline, which is FairMOT. BU-D, EG and TR stand for bottom-up decoupling, EG similarity matrix and tracking retrieval strategy, respectively. The best results are shown in **bold**.

Model Settings			Evaluation Indicators						
BU-D	EG	TR	IDF1↑	MOTA↑	HOTA↑	IDs↓	FP↓	FN↓	FPS↑
			75.6	71.1	-	327	-	-	-
✔			76.1	71.4	60.2	451	3319	11,655	19.7
✔	✔		78.1	72.5	61.5	186	3260	**11,430**	**24**
✔	✔	✔	**78.5**	**72.5**	**61.7**	**182**	**3212**	11,456	23.8

**Table 2 sensors-22-05863-t002:** Comparison of the state-of-the-art methods under the “private detector” protocol on the MOT17 test set. The best results are shown in **bold**. MOT17 contains rich scenes and half of the sequences are captured with camera motion. * indicates the addition of linear interpolation and † indicates JDE-based methods.

Method	HOTA↑	IDF1↑	MOTA↑	IDs↓	FP↓	FN↓	FPS↑
TraDes [30] †	52.7	63.9	69.1	3555	20,892	150,060	17.5
MAT [43]	53.8	63.1	69.5	2844	30,660	138,741	9.0
QuasiDense [16] †	53.9	66.3	68.7	3378	26,589	146,643	20.3
SOTMOT [44]	-	71.9	71.0	5184	39,537	118,983	16.0
TransCenter [45]	54.5	62.2	73.2	4614	23,112	123,738	1.0
GSDT [46] †	55.2	66.5	73.2	3891	26,397	120,666	4.9
PermaTrackPr [47]	55.5	68.9	73.8	3699	28,998	115,104	11.9
TransTrack [33]	54.1	63.5	75.2	3603	50,157	86,442	10.0
FUFET [28]	57.9	68.0	76.2	3237	32,796	98,475	6.8
FairMOT [12] †	59.3	72.3	73.7	3303	27,507	117,477	18.9
CSTrack [13] †	59.3	72.6	74.9	3567	23,847	114,303	15.8
Semi-TCL [48]	59.8	73.2	73.3	2790	22,944	124,980	-
ReMOT [49]	59.7	72.0	**77.0**	2853	33,204	**93,612**	1.8
CrowdTrack [50]	60.3	73.6	75.6	2544	25,950	109,101	-
CorrTracker [29] †	60.7	73.6	76.5	3369	29,808	99,510	15.6
RelationTrack [21] †	61.0	74.7	73.8	1374	27,999	118,623	8.5
SimpleTrack(Ours) †	61.0	75.7	74.1	1500	**17,379**	127,053	**22.53**
SimpleTrack(Ours) *	**61.6**	**76.3**	75.3	**1260**	22,317	116,010	-

**Table 3 sensors-22-05863-t003:** Comparison of the state-of-the-art methods under the “private detector” protocol on the MOT20 test set. The best results are shown in **bold**. The scenes in MOT20 are much more crowded than those in MOT17. * indicates the addition of linear interpolation and † indicates JDE-based methods.

Method	HOTA↑	IDF1↑	MOTA↑	IDs↓	FP↓	FN↓	FPS↑
MLT [51]	43.2	54.6	48.9	2187	45,660	216,803	3.7
FairMOT [12] †	54.6	67.3	61.8	5243	103,440	**88,901**	**13.2**
TransCenter [45]	-	50.4	61.9	4653	45,895	146,347	1.0
TransTrack [33]	48.5	59.4	65.0	3608	27,197	150,197	7.2
Semi-TCL [48]	55.3	70.1	65.2	4139	61,209	114,709	-
CorrTracker [29] †	-	69.1	65.2	5183	79,429	95,855	8.5
CSTrack [13] †	54.0	68.6	66.6	3196	25,404	144,358	4.5
GSDT [46] †	53.6	67.5	67.1	3131	31,913	135,409	0.9
SiamMOT [17] †	-	67.8	70.7	-	22,689	125,039	6.7
RelationTrack [21] †	56.5	70.5	67.2	4243	61,134	104,597	2.7
SOTMOT [44]	-	**71.4**	68.6	4209	57,064	101,154	8.5
SimpleTrack(Ours) †	56.6	69.6	70.6	2434	**18,400**	131,209	7.0
SimpleTrack(Ours) *	**57.6**	70.2	**72.6**	**1785**	25,515	114,463	-

**Table 4 sensors-22-05863-t004:** Data association comparison of different similarity matrices. The best results are shown in **bold**.

Similarity Matrix	IDF1↑	MOTA↑	HOTA↑	IDs↓	FP↓	FN↓	FPS↑
IoU	75.7	72.5	60.4	285	3510	11,048	**25**
GioU	66.4	70.4	54.8	378	4631	**10,956**	23.6
Embedding	64.1	65.0	53.4	749	6120	12,012	24.2
Embedding and Motion	76.1	71.4	60.2	451	3319	11,655	19.7
Embedding and IoU	77.2	72.3	61.4	263	**2560**	12,144	24
Embedding and GioU	**78.5**	**72.5**	**61.7**	**182**	3212	11,456	23.8

**Table 5 sensors-22-05863-t005:** Results of applying SimpleTrack to five different JDE-based trackers on the MOT17 validation set. Blue represents the tracking method using only the EG matrix, and red represents the tracking method combining the EG matrix and BYTE.

Method	Similarity	w/BYTE	HOTA↑	IDF1↑	MOTA↑	IDs↓	FPS↑
JDE [11]	EM	-	50.1	63.0	59.3	621	16.64
	EG	-	50.9	64.4	59.5	558	21.29
	EM	✔	50.4	64.1	60.2	437	18.52
	EG	✔	50.9	64.8	60.1	388	25.48
FairMOT [12]	EM	-	57.0	72.4	69.1	372	21.01
	EG	-	57.5	73.3	69.5	236	25.18
	EM	✔	-	74.2	70.4	232	-
	EG	✔	58.5	74.5	70.6	188	24.70
CSTrack [13]	EM	-	58.7	72.0	67.9	423	20.39
	EG	-	59.3	73.0	68.2	322	24.3
	EM	✔	59.8	73.9	69.2	298	20.72
	EG	✔	60.0	73.8	69.6	249	24.25
TraDes [30]	EM	-	58.6	71.7	68.3	293	15.8
	EM	✔	58.4	71.2	68.9	263	16.22
	EG	✔	59.0	71.5	68.5	483	16.5
QuasiDense [16]	EM	-	56.2	67.7	67.1	386	4.10
	EM	✔	58.5	71.9	67.4	295	4.80
	EG	✔	57.9	70.9	67.5	252	4.80

**Table 6 sensors-22-05863-t006:** Comparison of different association methods on the MOT17 validation set. JDE expresses the tracking strategy employed by [11,12,13,17] and BYTE expresses the tracking strategy employed by [10]. The best results are shown in **bold**.

Tracking Method	IDF1↑	MOTA↑	HOTA↑	IDs↓	FP↓	FN↓	FPS↑
JDE	76.1	71.4	60.2	451	3319	11,655	19.7
BYTE	75.7	72.5	60.4	285	3510	**11,048**	**25**
SimpleTrack (Ours)	**78.5**	**72.5**	**61.7**	**182**	**3212**	11,456	23.8

## Data Availability

The datasets in Experiment and Discussion are available at https://motchallenge.net.

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
