# Peer review of "SimpleTrack: Rethinking and Improving the JDE Approach for Multi-Object Tracking"

_sensors, 2022, doi:10.3390/s22155863_

Round 1

Reviewer 1 Report

Dear Authors,

   Many thanks for your manuscript submission to MDPI Journal of Sensors. This paper presented a multi-object tracking approach by reconsidering and improving the joint detection and embedding (JDE) based methods, which combined the embedding of Cosine distance and GioU distance of objects. The authors showed some qualitative and quantitative results to prove the merits of their approach in contrast to other methods. After careful review, I justify that this paper presents comprehensively good set of work, while several aspects still need improvement, which I enumerated as follows:

   a) Abstract: typically, the length of this section is within 150-200 words (hence, I think the current version is fine at length). But the current abstract missed to present some keynote quantitative results in the concluding remarks. Also, I think Lines 1-6 can be shortened, the rest narrations needs a bit more specific details, including added quantitative results in summary.

   b) Introduction: The Introduction section is a bit too generic, please consider doing a major rewrite for supplementing descriptions on the current research tasks and determining the points of your approach. If you citing more latest publications, they should be included in this section; meanwhile, I think the authors missed to present a summary of major contributions in the second last paragraph (typically 3-4 manifolds), and it is better to cascade a last paragraph for the organization on the remainder of this research article.

   c) Section 2: Related Work. While this section is quite concise, the main shortcomings on specification at this part is also explicit. The pros and cons on each of the three subtopics are not clear. Please consider rearranging the statements in this section, one suggested plan is to tabulate the crucial state-of-the arts in 2 tables for subsection 2.1 and 2.3. Thanks very much! 

   d) Section 3 (SimpleTrack Approach): I think the descriptions in this section is quite acceptable. While subsections 3.1 and 3.2 are clear, last subsection needs some improvement. Given that the pseudo code for the Algorithm takes over an entire page, please also consider applying some narrations on the main steps of the task-specific orientations for your approach. Besides, when you are presenting the algorithm, can you suggest some ways to improve the efficiency and evaluate the time cost of your algorithm? 

   e) Figures and Tables: Most figures are fine, while the inner characters inside Fig. 3 and Figs. 5-7 must be a bit larger. The tables contain sufficient quantitative results, and I think the numerical values of current results are fine. Again, be sure the size, resolution of each figure / table got calibrated to standard size (and 12 pt intervals before and after each figure or table).

   f) Experimental results: It is good to keep the quantitative analysis at the present version. The MOTA for Embedding at Table 2 should be 65.0. 

   g) Discussions: this paper missed a section on discussing the limitations of your study, the pros and cons of your proposed approach, and parallel comparsion to any latest publications (if reporting even better outcome other than your approach). Please consider filling this role if applicable. One suggestion is to split 4.3.1 ~ 4.3.8 into one subsection and shift the rest results to the discussion section, and apply with decent statements.

   h) Conclusions: This section had better be entitled with "Conclusions and Future Work". I think this paragraph need to be be organized with keynote quantitative scores on your concluding remarks, and be a bit more specific on what degree of improvement, what are the most crucial advantages of shortcomings on your proposed framework? Regarding future work, it is supposed to add a second paragraph, applying the specified summary of research challenges, and the orientations of prospective study should be a little bit more specific. Please re-arrange this section. Thanks a lot!

   i) References: Some problematic aspects should be addressed. (i) Apply the required abbreviated, italic formats on the title of journals when citing, i.e., "transactions" --> "Trans.". (ii) Please supplement any of missed information (volume and page numbers) for each conference proceedings and calibrate the citation style. (iii) I may recommend the authors proceed to review and cite both conventional, newer and latest approaches, in one decade range, especially these paper published in latest three years, i.e., Years 2019-2022 which are similar / parallel to your study, can be further enhanced in your upgraded version. These updates will make your citations look even more stronger. (iii) Please be sure comply with current MDPI template for other tutorial formats in a list of References at various sources.

   Some other minor issues needs your update are listed as follows:

   a) In some sections, the literal quality of English can be improved. I would recommend the peer-reviewed authors to polish the literal aspects of this research article, including grammatical checking and careful proofreading. 

   b) Apply uniform interval, font size and style on the characters of each figures, and fix the remaining formatting issues in the proofreading process.

   c) Please remove redundant half-space between sentences, and fix some minor typos in the context, i.e., in Line 129, figure 4 --> Figure 4; in Line 250, assocaition --> association, etc.

   Once again, wish you the best of luck for paper coming into acceptance. Thank you for your interests on publishing at MDPI Journal of Sensors. We look forward to reviewing your updated version for acceptance. Stay well!

With warm regards,

Yours sincerely,

Reviewer 2 Report

In this paper, the authors propose the Embedding and GioU matrix, that is actually a new association matrix, based on this matrix, they develop a simple and effective tracker for Re-identity and proposes a new tracking strategy based on our EG matrix. The authors have exhibited comprehensive experimental results, but there still have some issues and problems to be addressed.

For the structural of whole manuscript:

a.      In the introduction part, the author claims the proposed method and give us a simple background, the logical structural is coming out step-by-step, but at last they have not listed the contribution clearly. I think if you listed all the contribution point by point better at the last of this part.

b.     The Related works part, you just list some tasks related your work, but there are no analysis or comparisons on pros and cons. Please added some references and make some comparisons.

For the technical details:

a.      About the features, you have used JDE to extract many deep features, do these feature redundancy, please add some feature selection process.

b.     Decouple is important, in what way, you can verify the decoupling if better than other methods?

c.      In the ReId part, does cosine similarity the most effective way? How do you compared the similar feature by using cosine similarity? Layer-by-layer? Or in another way?

d.     How about the detection result for small objects and fast moving objects?

Reviewer 3 Report

In the manuscript some grammatical errors, such as “Kalman filiter” in line 132.

There are similar works that make use the JDE as a method to search for multiple objects:

- Mouawad, I., & Odone, F. (2022). Faster Video: Efficient Online Joint Object Detection and Tracking. In International Conference on Image Analysis and Processing (pp. 375-387). Springer, Cham.

- Meinhardt, T., Kirillov, A., Leal-Taixe, L., & Feichtenhofer, C. (2022). Trackformer: Multi-object tracking with transformers. In Proceedings of the IEEE/CVF Conference on Computer Vision and Pattern Recognition (pp. 8844-8854).

  - Chan, S., Jia, Y., Zhou, X., Bai, C., Chen, S., & Zhang, X. (2022). Online Multiple Object Tracking Using Joint Detection and Embedding Network. Pattern Recognition, 108793.

What would be the difference between your proposed method, with that of the previous references?

Reviewer 4 Report

The paper describes improvements to existing multi-object solutions by utilizing by introducing incremental steps to existing frameworks. The overall approach is strongly based on deep learning pipelines and follows the general approach of tracking-by-detection. Overall, the paper is well written and easy to follow. The content is strictly focused on joint detection and embedding methods.

Towards this end several modifications of existing frameworks are described leading to an overall incremental progress. The evaluation, as well as the ablation station are sound, although some of the results appear to be marginal.

As a slight drawback the paper follows the overall mainstream of benchmarking publications alongside with its shortcomings. Therefore, therefore the soundness of introduced building blocks are evaluated solely on the used dataset which leaves the question open on how general this approach will be in other domains, including robustness and the amount of training data needed. Further, there will be no insight into the mechanics of the network, although the author give some statements about the feature selection behavior after training.

In my opinion the paper will give a solid conference publication, while it may be a little bit too weak for a journal publication.
